# Estimating the impact of COVID-19 vaccine inequities: a modeling study

Nicolò Gozzi[1,2], Matteo Chinazzi[3], Natalie E. Dean [4], Ira M. Longini Jr[5], M. Elizabeth Halloran [6,7], Nicola Perra [3,8] ✉ & Alessandro Vespignani[3]

Access to COVID-19 vaccines on the global scale has been drastically hindered by structural socio-economic disparities. Here, we develop a data-driven, age-stratified epidemic model to evaluate the effects of COVID-19 vaccine inequities in twenty lower middle and low income countries (LMIC) selected from all WHO regions. We investigate and quantify the potential effects of higher or earlier doses availability. In doing so, we focus on the crucial initial months of vaccine distribution and administration, exploring counterfactual scenarios where we assume the same per capita daily vaccination rate reported in selected high income countries. We estimate that more than 50% of deaths (min-max range: [54−94%]) that occurred in the analyzed countries could have been averted. We further consider scenarios where LMIC had similarly early access to vaccine doses as high income countries. Even without increasing the number of doses, we estimate an important fraction of deaths (min-max range: [6−50%]) could have been averted. In the absence of the availability of high-income countries, the model suggests that additional non-pharmaceutical interventions inducing a considerable relative decrease of transmissibility (min-max range: [15−70%]) would have been required to offset the lack of vaccines. Overall, our results quantify the negative impacts of vaccine inequities and underscore the need for intensified global efforts devoted to provide faster access to vaccine programs in low and lower-middle-income countries.

Throughout the COVID-19 pandemic, socio-economic disparities have been linked to higher and disproportionate COVID-19 burden, a finding replicated across different countries and social strata[1–10]. In this context, structural inequities in access to COVID-19 vaccines were discussed even before any specific vaccine was authorized by regulatory agencies[11–15]. Despite international initiatives for equitable sharing agreements such as the COVID-19 Global Vaccine Access (COVAX) program[16,17], vaccine nationalism has largely superseded global equity efforts. Indeed, the differences in terms of COVID-19 vaccines doses, administered across countries grouped by income levels, are staggering[18–21]. These inequities have potentially enormous effects on the economies and future health of lower middle and low income countries (LMIC)[11,12,22–25]. However, except for three recent studies that modelled the effects of different global vaccine allocation and sharing strategies[26–28], a quantitative, detailed, and tailored estimation of the consequences of vaccine inequity across several LMIC is largely missing.

Here, we develop a stochastic, multi-strain compartmental epidemic model applied to twenty LMIC selected from all WHO regions.

[1]Networks and Urban Systems Centre, University of Greenwich, London, UK. [2]ISI Foundation, Turin, Italy. [3]Laboratory for the Modeling of Biological and Socio-technical Systems, Northeastern University, Boston, MA, USA. [4]Department of Biostatistics and Bioinformatics, Emory University, Atlanta, GA, USA. [5]Department of Biostatistics, College of Public Health and Health Professions, University of Florida, Gainesville, FL, USA. [6]Fred Hutchinson Cancer Center, Seattle, WA, USA. [7]Department of Biostatistics, University of Washington, Seattle, WA, USA. [8]School of Mathematical Sciences, Queen Mary University, London, UK. ✉e-mail: n.perra@qmul.ac.uk

The epidemic model accounts for national demographics, age-structured contact mixing patterns, the impacts of non-pharmaceutical interventions (NPIs), multiple virus strains and their variable effects on vaccines' efficacy. We fit the model to each country independently using an Approximate Bayesian Computation method based on Sequential Monte Carlo[29,30] and we explore a range of counterfactual scenarios aimed at quantifying the effect of a higher or earlier vaccine availability on COVID-19 mortality as of 2021/10/01 in the twenty LMIC considered. The results suggest that, if these countries could have afforded the same per capita daily vaccination rate reported in selected high income countries, more than 50% of deaths (min-max range: [54−94%]) that occurred could have been averted. Even without more doses, if these countries had a similar early access to vaccine doses as high income countries, an important fraction of deaths (min-max range: [6−50%]) could have been averted. We also estimate the level of NPIs that these countries would have needed to offset the lack of vaccines. Indeed, while in the first phases of the emergency the mitigation of the pandemic was achieved, around the world, at high costs through the implementation of economically and socially disruptive NPIs[15], in high-income countries vaccines have facilitated relaxing such tough socio-economic measures[31,32]. We find that significantly more effective or prolonged NPIs would be needed to observe the same number of averted deaths estimated in high vaccine availability scenarios.

In summary, our findings emphasize the negative consequences of inequities in the access to COVID-19 vaccines and advocate for concerted efforts to expedite and ensure fair distribution of vaccines in LMICs. This is not just a moral imperative to reduce the burden of COVID-19 around the world, but also a practical stance to limit the emergence, spread, and introduction of new variants possibly able to breach the protection of existing vaccines. Though we have focused on twenty LMIC, the approach we developed could be used to study and quantify the impact of inequitable vaccination in other countries.

## Results

### Quantifying vaccine inequities

To quantify vaccine access and administration across countries with different income levels, we combine two datasets (see the Methods section and the Supplementary Information for more details). The first one is collected and updated by the United Nations Development Programme via their Global Futures Platform[18]. It provides general information about COVID-19 vaccine data around the world, including several socio-economic dimensions. The second dataset, made available by Our World in Data[33], provides a range of general information about country-level vaccination efforts, including the number of doses administered as a function of time which we have used as one of the inputs in the modelling effort described below.

As of October 1st 2022, 77% of individuals living in high and upper middle income countries completed the initial COVID-19 vaccination course (i.e., one or two doses depending on the vaccine). The equivalent share in LMIC was 50%, around 1.5 times lower. Inequities were drastically more pronounced during the earliest stages of the vaccination rollout, when populations also had much lower levels of infection-induced immunity. In Fig. 1a, we plot the total number of doses administered per 100 people. By October 1st, 2021 (i.e., the horizon of our analysis), high and upper middle income countries had, on average, more than one dose per person. The numbers are drastically different in LMIC. While lower middle income countries

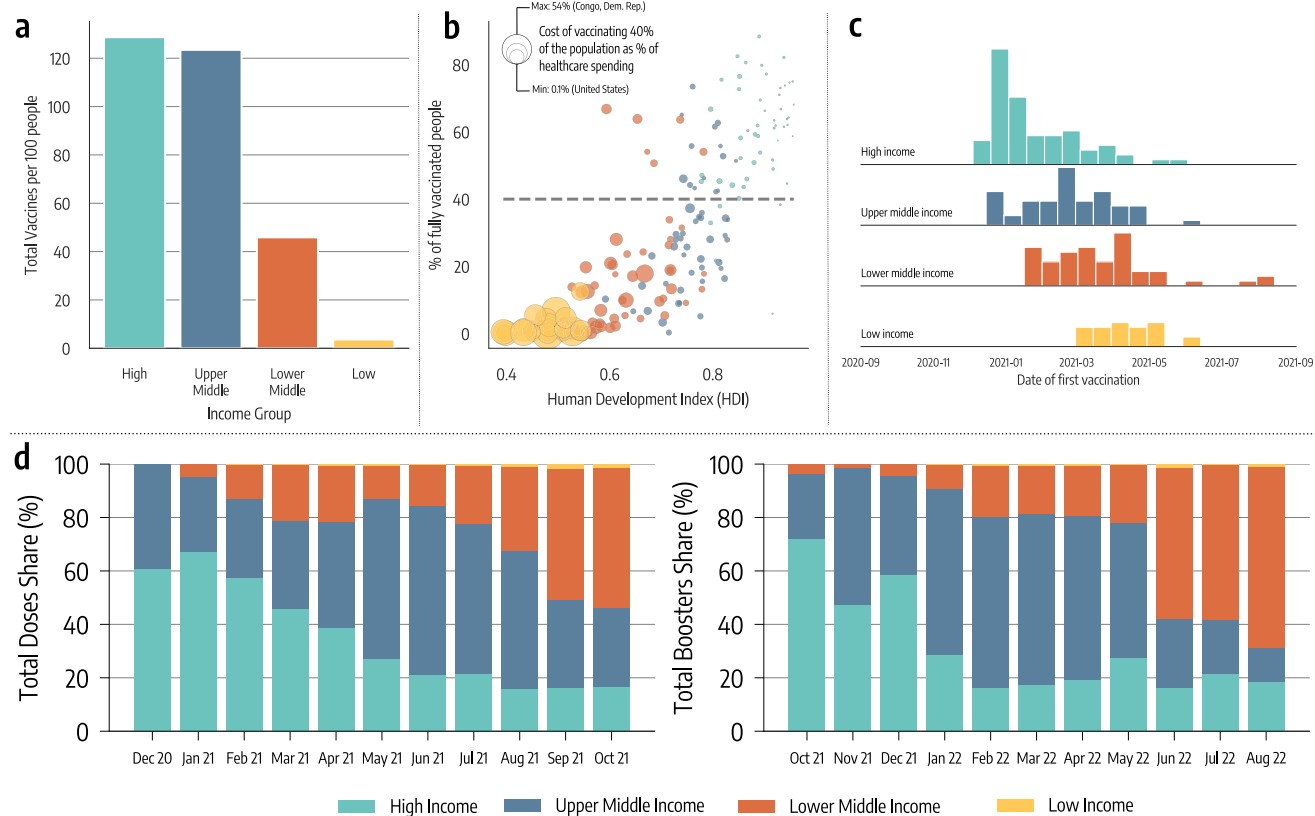

**Fig. 1 | Vaccine inequities. a** Total number of doses administered per 100 people in different income groups as of October 1, 2021. **b** Scatter plot of % of a country's population who is fully vaccinated versus their Human Development Index (HDI). The color of dots indicates the country's income group while size is proportional to the cost of vaccinating 40% of the population as a percentage of current healthcare spending. **c** Histograms of the date of first COVID-19 vaccination across different country income groups. **d** Evolution in the share of doses administered monthly across country income groups (left hand), and evolution of monthly booster doses share (right hand side).

administered slightly more than 40 doses per 100, the equivalent number for low income countries is only 3.6 doses per 100.

In Fig. 1b, we plot the percentage of each country's population that was fully vaccinated by October 1st, 2021, versus the country's Human Development Index (HDI). The HDI is a composite index that accounts for life expectancy, education, and per capita income as well as other aspects of human development[34]. The size of the data points is set proportional to the estimated cost of vaccinating 40% of the population as a percentage of the country's current healthcare spending; this metric is used to quantify the economic challenge posed by achieving the 40% vaccination level proposed by WHO as an interim target by the end of 2021[18]. The plot shows a strong positive correlation between HDI and vaccination coverage (Pearson correlation coefficient: 0.79, $p < 0.001$). The more developed the country, the higher the fraction of its population vaccinated. Furthermore, countries characterized by the lowest values of HDI face drastically bigger economic challenges in reaching 40% of the population vaccinated. Unfortunately, even one year later, as of October 1st 2022, only half of lower middle income countries have vaccinated at least 40% of the population, while only one low income country out of ten met the target.

In Fig. 1c, we plot the distribution of the vaccine administration starting dates by country income group. For high and upper middle income countries this date is generally much earlier than for LMIC. This figure clearly highlights the different logistic challenges in setting up a mass vaccination campaign across countries and speaks to the differential power among income levels to secure a scarce resource such as COVID-19 vaccines in the early phases of the rollout[19].

In Fig. 1d, we plot the share of vaccines administered globally across country income levels by month, starting in December 2020. The plot confirms that, in the first seven months of the COVID-19 vaccination campaign, more than 80% of the doses were concentrated in high and upper middle countries. The level of inequity becomes even more staggering when considering the share of the global population of the four groups. Only 16% of the global population lives in high income countries, while nearly 50% live in LMIC (8% in low income countries). Hence, the plot highlights how, as of the 1st of October 2021, low income countries had administered a share of doses that is smaller than 1% of total doses. As shown on the right hand side of Fig. 1d, high income countries similarly monopolized the administration of booster doses in late 2021 and the first half of 2022, when most LMIC were still far behind with primary vaccinations.

## Modeling the vaccination campaigns in LMIC

To quantify the impact of vaccines inequities on COVID-19 burden, we developed a stochastic and multi-strain epidemic model. We consider a SEIR-like natural history of the disease, and the model takes as input for each country its demographics, proxy data for non-pharmaceutical interventions (NPIs), age-stratified contact matrices from Ref. 35, variants prevalence, and epidemic data describing confirmed deaths. Vaccine administration is explicitly modeled with the number of daily 1st and 2nd doses in different countries from Ref. 36. The model is stochastic and transitions among compartments are simulated through chain binomial processes. We consider individuals who received one or two vaccine doses, with vaccine efficacy against infection and death for one and two doses and as a function of the specific circulating variant of the virus. We also assume that vaccinated individuals who get infected are less infectious by a factor accounting for the forward transmission reduction observed in vaccinated people[37]. Details of the mathematical definition and computational implementation of the model are summarized in the Methods section and fully described in the Supplementary Information.

In Fig. 2a, we map the geographical location of the twenty LMIC covering all six WHO regions that were selected for modeling. The selection process was driven by data availability about vaccinations, genomic information about variants prevalence, access to NPIs proxy

data, and reported deaths. Across the selected countries, we capture a wide range of vaccination coverage. By October 1, 2021, more than 50% of the populations in Sri Lanka, El Salvador, and Morocco had completed the initial COVID-19 vaccination course, compared to fewer than 3% in Ghana, Uganda, and Zambia. Countries such as the Philippines, Indonesia, Honduras, and Bolivia are in the middle with fractions between 19% and 24%. We calibrate the model in each country separately via an Approximate Bayesian Computation method based on Sequential Monte Carlo[29,30] using as evidence the time series of recorded deaths in each country. This approach allows us to find the posterior distributions for a range of parameters such as the effective transmissibility of the different strains, seasonality, delay between deaths and their notification, under-reporting of deaths, and infection fatality rates (IFRs)[38] in the different countries as reported in Methods section. We first use the model to estimate, for each country, the number of deaths averted by the actual vaccination campaign, relative to a setting where no vaccines were available (results reported in the Supplementary Information). The model is run in the period 2020/10/01−2021/10/01, covering one year of vaccine allocation and distribution, prior to the emergence of the Omicron variant. We refer the reader to the Supplementary Information for details about the calibration process and an estimate of the impact of the actual vaccination campaigns in averting deaths.

## Counterfactual vaccination scenarios

We use the model to study counterfactual scenarios in which (i) the per capita vaccination rate in the selected countries would have equaled that of high-income countries, and (ii) vaccination rollout starts at the same time as in high-income countries, with no change in overall volume. In administering the extra available doses we assume a protocol that prioritizes the elderly population hence targeting a reduction of deaths rather than the reduction of the overall infection incidence[39–41]. Our aim is to quantify the untapped benefits of vaccines in LMIC rather than to study alternative global strategies of allocation (as done in Refs. 26–28). Hence, we assume a higher (first scenario) or an earlier (second scenario) availability of doses. For each counterfactual scenario, we estimate the impact of the changed vaccine rollout as the percent reduction in deaths (averted deaths) during the simulation period (2020/10/01−2021/10/01) compared to the actual vaccine rollout (see the Supplementary Information for more details). We note how the scenario without vaccines could be used as an alternative baseline. The goal is to quantify the impact of vaccine inequities, hence we opted to select the factual rollout as the baseline.

In Fig. 2b, we report the estimated percentage of deaths averted and the median absolute numbers of deaths averted per country assuming the first counterfactual scenario of a US-equivalent vaccination rate. For more than half of the countries, the percentage of deaths averted exceeds 70%, with peaks above 90% for Afghanistan, and Uganda. In these countries, the absolute numbers of averted deaths are staggering, ranging from 149,000 (IQR: [122,000−183,000]) in Indonesia to 1700 (IQR: [1100−2600]) in Rwanda. For the other half of the countries, the percentages of deaths averted are in the 50−70% range, with absolute numbers ranging from 20,600 (IQR: [15,400−26,800]) for Mozambique to 2200 (IQR: [1700−2700]) in El Salvador. In the Supplementary Information we show how analogous results can be obtained considering vaccination rates of other high-income countries such as the European Union (EU) and Israel. Indeed, in both cases we observe a drastic reduction of fatalities. When considering dose availability of Israel the averted deaths in LMIC span between 60% to nearly 100%, and from 40% to 90% when EU rates are considered.

The second counterfactual scenario investigates the impact of an earlier start of the vaccination campaigns without changes in allocation volume. In other words, this scenario considers a more equal timing in the allocation of vaccine doses. We take as a reference point

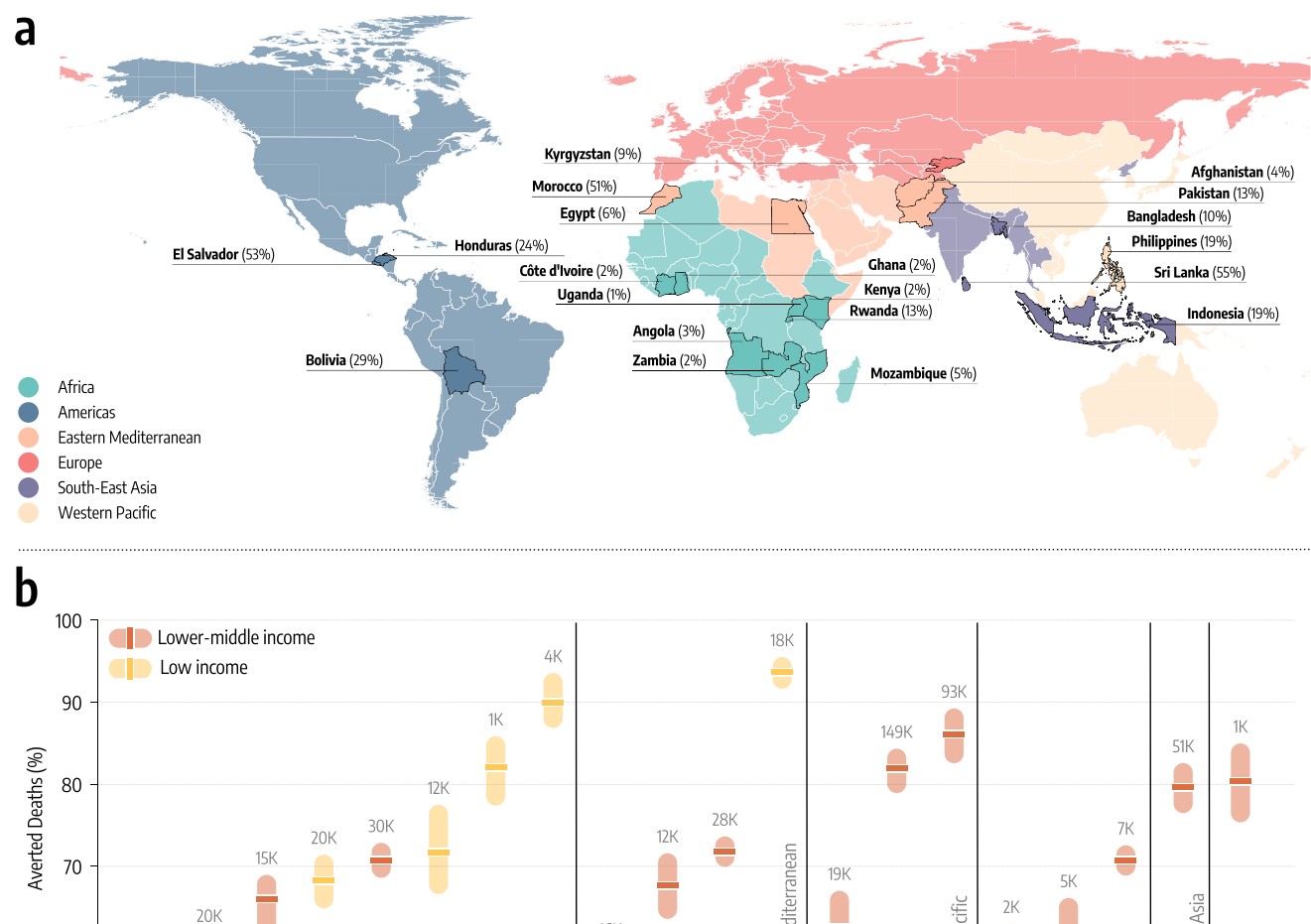

**Fig. 2 | Counterfactual scenarios - Deaths averted if countries had US-equivalent vaccination rate. a** Countries modeled, their WHO region, and the percentage of fully vaccinated individuals there as of October 1, 2021. World borders layer is taken from Ref. 70. **b** Deaths averted expressed as a percentage with respect to the actual vaccination rollout (median and interquartile range computed over 1000 independent model realizations), assuming per capita vaccination rates equivalent to the United States. Averted deaths are computed over the simulation period (2020/10/01–2021/10/01). The median absolute number of deaths averted is reported above the inter-quartile range.

the start of the vaccination campaign in the US which took place on December 14, 2020. Among the countries under study, Indonesia administered the first doses in mid-January 2021, and Bangladesh, Egypt and Sri Lanka in late-January 2021. Eight countries started their vaccination in February: Bolivia and Pakistan in the beginning, El Salvador, Morocco, and Rwanda in the middle, Afghanistan, Honduras, and Philippines at the end of the month. Seven countries started their campaigns only in March: Angola, Côte d'Ivoire, Ghana, Mozambique and Uganda within the first ten days of the month while Kyrgyzstan at the end of it. Finally, only one of the twenty countries under investigation (Zambia) started vaccinations in April (mid month). Hence, the delay with respect to the US starting date spans one to four months. Figure 3 shows how an earlier start would have been beneficial as it would have found a larger fraction of the population susceptible before the Delta wave. However, the magnitude of such effects is heterogeneous and overall smaller when compared to the previous counterfactual scenario with more vaccine doses. A key factor is the interplay between the amount of vaccine available and the relative shift of the starting time. Sri Lanka, El Salvador and Morocco, that achieved

the highest coverage in the group and initiated their campaigns around two months later compared to the US, would have averted more than 40% (50%, IQR: [45−56%] for Morocco) of deaths. In the case of Angola and Zambia, both countries with a very low vaccination rate, a three/four month head start makes a difference (6400, IQR: [5000−8600] and 4000 [3100−4800] averted deaths respectively), but its effects are ultimately diminished by the relatively small number of administered vaccine doses. In fact, the percentage of averted deaths is very similar to what we find for Pakistan that had a higher vaccination coverage but started only about two months later than the US. Similarly, the moderate percentage of averted deaths for Indonesia and Bangladesh, which however would have resulted in 38,000 (IQR: [31,500−45,700]) and 22,000 [15,300−31,300] fewer deaths, is due to the small difference between the actual and counterfactual start of the vaccination campaigns.

## Estimate of NPIs required to offset vaccine inequity
In the counterfactual scenarios studied above, our modeling approach considers the same level of NPIs implemented in the factual scenario.

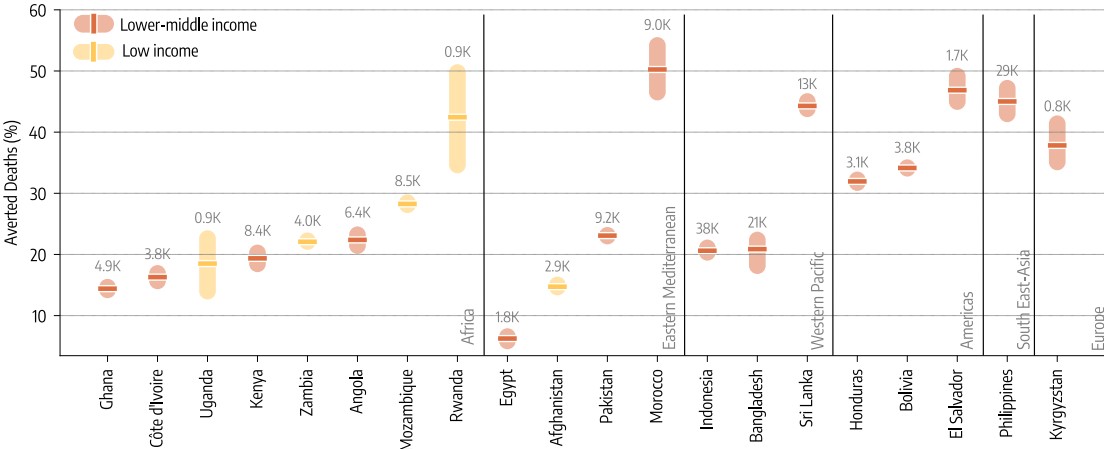

**Fig. 3 | Counterfactual scenarios - Deaths averted if countries had US-equivalent vaccination start date.** Deaths averted expressed as a percentage with respect to the actual vaccination rollout (median and interquartile range computed over 1000 independent model realizations), assuming United States start date of December 14, 2020. Averted deaths are computed over the simulation period (2020/10/01–2021/10/01). The median absolute number of deaths averted is reported above the inter-quartile range.

For this reason, we have investigated the extent of additional NPIs that would be needed to offset the limited vaccine availability in LMIC. In this third counterfactual scenario, we keep the doses administration as it unfolded in reality. Then, at week 51 of 2020, the start of the vaccination campaign in the US, we modify the NPIs to make them more restrictive thus reducing the transmissibility. Since the impact of NPIs is modulated by their strictness and duration[15,42–47], we explore a two-dimensional phase space in which additional NPIs are introduced for $W$ weeks (after week 51 of 2020) and decrease transmissibility by $X$%. It is important to note how the strengthening of NPIs could be achieved via further social distancing in concert with less disruptive measures like wider face mask adoption (see Supplementary information for the modeling implementation). In Fig. 4 we show the percentage decrease in transmissibility that, if achieved for four months, would avert the equivalent numbers of deaths as achieved by a US-level vaccination rate. Notably, nearly half of the countries would have needed reductions of 40% or more during that period. The other half would need reductions of around 20–30%. These variations to the transmissibility could be hardly achieved without a significant strengthening of social distancing measures. Less disruptive policies, such as face masks, could help but their contribution is strongly dependent on their adoption and effectiveness[48,49]. Côte d'Ivoire and Angola are the countries requiring the strongest additional NPIs to match the benefit of higher vaccination rates. As of October 1st, 2021, they vaccinated only 2% and 3% of their population respectively. Although one might think that the key driver of the trend is vaccination coverage, Pakistan, that in the same period vaccinated 13% of the people, is among the countries requiring the least additional NPIs to match the benefit of higher vaccination rates. Pakistan experienced a large epidemic wave in May 2021 and managed a rapid start of the vaccination campaign. Honduras (which ranked third in terms of additional NPIs needed) instead started its campaign only a few weeks later but the initial vaccination rate was very slow (see the Supplementary Information for details). Hence, the starting date and the rate of vaccination are equally important in defining the effectiveness of vaccination campaigns. Furthermore, in this counterfactual scenario, additional NPIs are implemented in different epidemiological contexts. The impact of NPIs has been critically linked to their timing in reference to the disease progression[15,42–47]. For instance, Ghana experienced a big wave at the beginning of 2021, while Kyrgyzstan and Uganda, which are among the countries needing the least additional NPIs to match the effects of higher vaccination rate, experienced big

waves later, only in the summer of 2021. In Fig. 4b we show a phase diagram for Pakistan, Philippines, and Ghana which are the countries respectively in the bottom, middle, and top rank according to the percentage decrease in transmissibility needed to offset vaccine inequities. The Figure details the averted deaths (color scale) as a function of NPIs duration (x-axis) and decrease in transmissibility (y-axis). Overall, the plot shows that the longer additional NPIs are in place, the less strict they need to be to avert the same number of deaths. Furthermore, Pakistan would have required a decrease in transmissibility of 30% for two months to match the benefit brought by a higher vaccination rate. In contrast, in the case of the Philippines and Ghana, we would need a 45% and 90% decrease in the same period. These observations highlight how the effectiveness of the vaccination campaigns is affected by the interplay of their start, rate, coverage, and timing in reference to the disease progression.

## Discussion

Global COVID-19 vaccines allocation has been characterized by extreme inequities. As a result, high and upper-middle income countries managed vaccination rates and coverage that are much higher than LMIC. Furthermore, their rollout started earlier, and it had a much smaller economic impact (with respect to their GDP and healthcare expenditures). In this context, we studied the impact of COVID-19 vaccine inequities in twenty LMIC selected from all WHO regions by quantifying the potential effects of higher or earlier doses availability. We developed an epidemic model, calibrated to the epidemiological context of each country, to study counterfactual scenarios where the per capita vaccination rate in LMIC would have equaled that of high-income countries. We found that the twenty countries would have averted more than half (with peaks above 90%) of the deaths that actually occurred. We also ran a counterfactual scenario assuming a vaccination rollout start at the same time as in high-income countries, with no change in allocation volume. Also in this case, we found that a significant fraction of deaths would have been averted. Finally, we estimated the strictness of NPIs that each country would have to put in place to offset the lack of vaccines with respect to high-income countries. Across the different LMIC, we found that stronger and sustained NPIs would have been necessary. This result, combined with the difficulty of implementing additional NPIs in these settings, underlines the largely untapped benefits that vaccines could have brought to LMIC. Overall, our findings are in line with those reported in the still small literature focused on COVID-19 vaccine inequities[26–28].

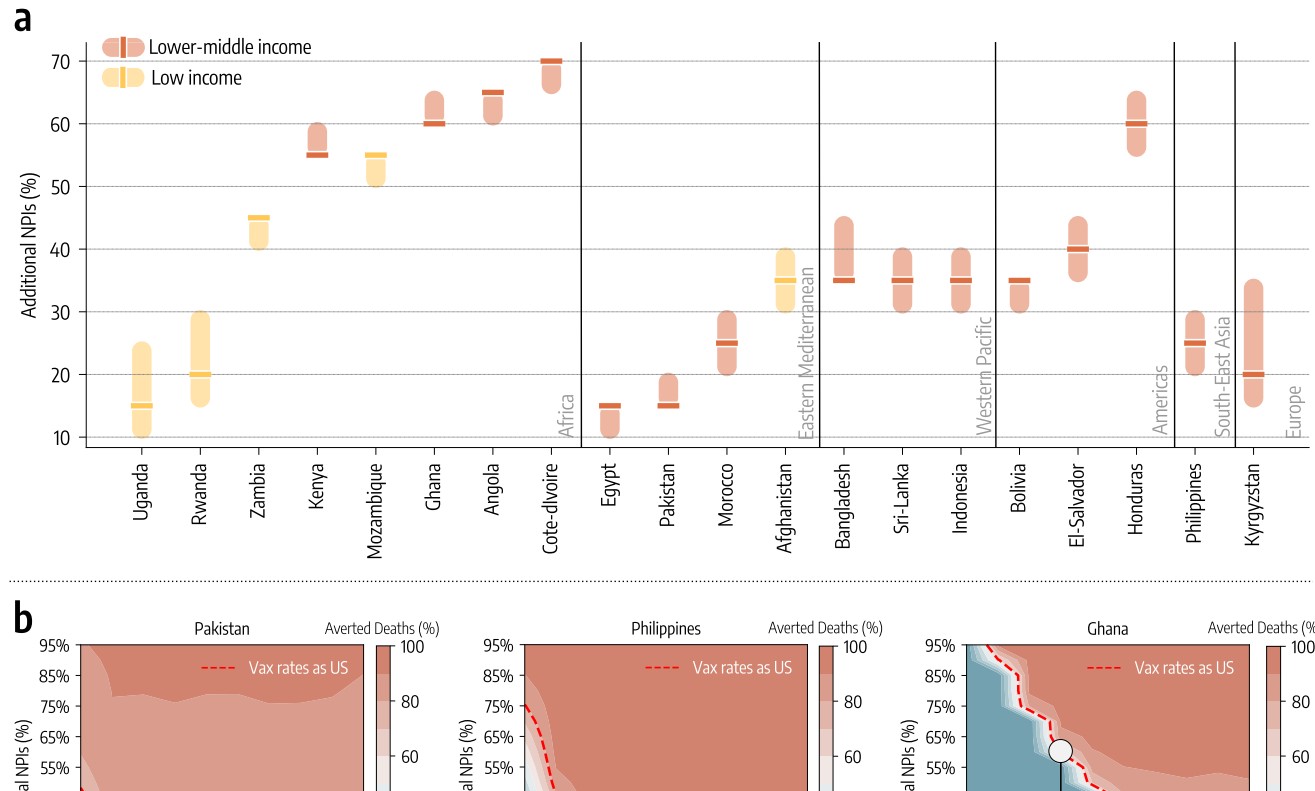

**Fig. 4 | The role of NPIs. a** Additional decrease of transmission obtained through stricter NPIs, put in place for four months, needed to match the deaths averted that the vaccination rate of the US would have allowed (median and interquartile range computed over 1000 independent model realizations). **b** For three countries we show the contour plots of the percentage of deaths averted (median %) with stricter and/or longer NPIs, relative to the actual vaccination baseline. Percentage of deaths averted achieved by a US-equivalent vaccination rate is plotted as reference (red dashed line). The white circle indicates the level corresponding to stricter NPIs sustained for additional 16 weeks.

Though the differences in the counterfactual scenarios considered make quantitative comparisons difficult, previous results confirm the negative effects of vaccine nationalism and of allocation based on income rather than need[26–28].

Our approach comes with the limitations of modeling studies. The details about the vaccination campaigns in some of the countries are limited. For example, information about the types of vaccines administered in some cases is missing. The model operates at a national level thus neglecting geographical heterogeneities that could reveal yet other layers of inequities, within each country. Although we consider variable IFRs across countries, we do not explicitly account for comorbidities or limited healthcare access. The model includes the under-reporting of deaths as a parameter fitted in the whole period under investigation. Hence, temporal variability in the ascertainment of deaths is neglected. To address this point, in the Supplementary Information we present a second model including a time-varying parameter. The main findings are not affected by this change in the modeling setup. The transmission rate in each country is fitted, thus allowing to capture differences in behaviours and measures that might affect transmissibility. However, we do not explicitly model protection offered by face masks or by increased hygiene measures. We do not investigate other counterfactual scenarios based on changes in vaccination protocols, such as for example delaying the time between doses. This strategy, which aims to prioritize one-dose protection

coverage, has been shown to be effective in the first ten months of the rollout in the United Kingdom[50]. Finally, the counterfactual scenarios do not consider changes in the global allocation of doses, the real global availability of vaccines, nor the local cost of the supply chain necessary to receive, store, distribute and administer doses.

The overall picture emerging from our analysis shows that vaccine inequity, in both the number of doses available and the timeline of delivery, drastically reduced the impact of vaccination campaigns in the group of LMIC studied. The emergence of new variants, featuring higher transmissibility and immune escape, suggests how vaccine inequity might become even more critical as we move forward in the pandemic[51]. Indeed, the original COVID-19 vaccines have been challenged by new variants (such as Omicron) when it comes to protection from infections and mild disease, but they still offer a very significant protection from severe outcomes. Progress in vaccine availability and administration in LMIC, also with respect to the administration of booster doses, would help considerably to mitigate the risks associated with highly transmissible new SARS-CoV-2 lineages.

While our results do not account for the constraints in the number of globally available doses, estimates suggest that hundreds of millions of vaccines have gone wasted in the period of time we considered, rapidly reaching one billion by July 2022[52]. Vaccine hesitancy, expiration dates, logistic issues, and vaccine nationalism all played a role, especially in high-income countries[53,54]. Moreover, the overall number

of doses available at any given time is influenced by many variables such as production choices, supply chain capabilities, vaccination protocols (e.g., the time between first and second doses)[50], and national/international policies among others. The choices made to affect these variables evolved over time, clearly showing that alternative models are possible. The presented approach and results can be extended to other countries and are potentially relevant in defining strategies aimed at minimizing the effect of inequities in vaccine allocation across countries.

## Methods

### Data

Data on global vaccine inequities come from the United Nations Development Programme via their Global Futures Platform[18]. The data detailing the timeline of vaccinations, used in the simulations, come from Our World in Data[33]. The dataset provides the cumulative share of people partially and fully vaccinated against SARS-CoV-2 as a function of time.

Data about demographics come from the United Nation World Population Prospects[55]. Epidemiological data are extracted from the COVID-19 Data Repository by the Center for Systems Science and Engineering (CSSE) at Johns Hopkins University and from official sources[56]. Data to estimate the impact of NPIs on transmission dynamics come from the COVID-19 Community Mobility Report By Google[57]. The dataset provides the percentage change in mobility $r(t)$ on day $t$. We compute $r(t)$ by using the average of the fields *workplaces percent change from baseline*, *retail and recreation percent change from baseline* and *transit stations percent change from baseline*. We refer the reader to the Supplementary Information for more details.

### Epidemic model

We adopt a SEIR-like stochastic compartmental model. Susceptible individuals are placed in the compartment $S$, by getting in contact with the Infectious ($I$) they transition to the compartment of the Exposed ($E$). Exposed individuals are infected and transition to the compartment $I$ with rate $\epsilon$. Infectious subjects leave the compartment with rate $\mu$. Following the literature describing the COVID-19 characteristics[58,59], we set $\epsilon = 1/4$ days$^{-1}$ and $\mu = 1/2.5$ days$^{-1}$. We compute the number of deaths from the daily recovered as follows. Individuals that exit from the $I$ compartment, can either transition to the Recovered ($R$) or the Dead ($D$) compartment. The share of individuals transitioning to the $D$ compartment is regulated by the age-stratified Infection Fatality Rate (IFR) from Ref. 38. To account for delays due to hospitalization and reporting, we record the number of deaths computed on the recovered of day $t$ only after $\Delta$ days. Hence, $D$ individuals transition to the compartment $D^o$ (superscript $o$ stands for "observed") at a rate $1/\Delta$. Individuals are divided into 10 age groups (0–9, 10–19, 20–24, 25–29, 30–39, 40–49, 50–59, 60–69, 70–79, 80+). The age-stratified rates of interaction are defined by the country specific contacts matrix C from Ref. 35. The model includes a seasonal term to capture modulation of the force of infection regulated by changes in factors such as temperature and humidity (see more details in the Supplementary Information)[60,61].

The model accounts for vaccinations. We assume that all individuals except the infectious can receive the vaccine. The per-capita rate at which susceptible individuals, that received a dose of vaccine, get infected (i.e., force of infection) is reduced by a factor $(1 - VE_{S1})$. If these individuals get infected, their IFR is also reduced by a factor $1 - VE_{M1}$. Hence, the overall efficacy of a single dose of vaccine against death is $VE_1 = 1 - (1 - VE_{S1})(1 - VE_{M1})$. The force of infection for susceptible individuals that received two doses, and the IFR are reduced, respectively, by $(1 - VE_{S2})$ and $(1 - VE_{M2})$, implying an overall efficacy of $VE_2 = 1 - (1 - VE_{S2})(1 - VE_{M2})$. We also assume that vaccinated individuals that get infected are less infectious by a factor $(1 - VE_I)$[37]. Since vaccine protection is not immediate, we introduce a delay of $\Delta_V$ days between administration (of both 1st and 2nd dose) and the actual effect of the vaccine. For example, an individual who received the 1st dose on day $t$, will be protected with efficacy $VE_1$ only, on average, after $\Delta_V$ days. We set $\Delta_V = 14$ days. As we do not have detailed information about the age of individuals receiving vaccines in all the countries considered, we assume that the rollout proceeds prioritizing the elderly. This is the strategy followed by the vast majority of governments worldwide[40,62,63]. Vaccines are distributed in decreasing age order until all 50+ individuals are vaccinated, after vaccines are distributed homogeneously to the age groups 10−50. We inform the model with the number of daily 1st and 2nd doses in different countries from Ref. 36. We set $VE_1 = 80\%$ ($VE_{S1} = 70\%$), $VE_2 = 90\%$ ($VE_{S2} = 80\%$), and $VE_I = 40\%$[37].

We add specific $E$ and $I$ compartments to account for the introduction and emergence of a variant of concern. Considering the period under examination and the evidence from genomic surveillance in all countries under examination we consider the arrival and spread of the SARS-CoV-2 variant of concern Delta. Looking at genomic sequence data from Ref. 64−66 we get a proxy date for its introduction (see more details in the Supplementary Information). We assume that Delta is $\psi$ times more transmissible than the strain circulating previously and has a shorter latent period $\epsilon_{Delta}^{-1} = 3$ days [67]. We also assume that vaccines have a reduced efficacy against Delta VOC: $VE_1^{Delta} = 70\%$ ($VE_{S1}^{Delta} = 30\%$), $VE_2^{Delta} = 90\%$ ($VE_{S2}^{Delta} = 60\%$)[37].

The model accounts also for the effects of non-pharmaceutical interventions (NPIs) on transmission. To this end, we adopt as a proxy the COVID-19 Community Mobility Report By Google[57]. This data provides the percentage reduction of individuals visiting specific locations on a given day. Here, we derive a single mobility reduction parameter $r(t)$ by averaging the different fields of the report, and we convert it into a contacts reduction parameters $c(t)$ following the relation: $c(t) = (1 + r(t)/100)^2$. Indeed, under a homogeneous mixing assumption the number of contacts scale with the square of the number of individuals. To account for the modulation in contacts induced by NPIs in the simulations, the contact matrix C is multiplied by this reduction parameter $c(t)$. We refer the reader to the Supporting Information for more details about the modeling framework.

### Model calibration

The free parameters of the model are calibrated through an Approximate Bayesian Computation based on Sequential Monte Carlo (ABC-SMC)[29,30]. The free parameters investigated are:

- the transmission rate $\beta$; we explore uniformly values such that the reproductive number $R_t$ on the first simulation date is between 0.6 and 2.0;
- the delay in deaths $\Delta \sim U(10, 35)$[68];
- the seasonality parameter $\alpha_{min} \sim U(0.5, 1.0)$ (0.5 indicates strong seasonality while 1.0 absence of seasonality);
- the initial number of infected individuals; we explore uniformly values between 1 and 1000 times the number of cases notified in the 7 days prior the beginning of the simulation ($Inf_{start}^{mult}$). We divide these individuals in the infected compartments ($L$, $I$) proportionally to the time spent there by individuals ($\epsilon^{-1}$ for $L$ and $\mu^{-1}$ for $I$);
- the initial number of recovered; we explore uniformly values between 1 and 100 times the total number of reported cases up to the start of the simulation ($Rec_{start}^{mult}$);
- the relative transmissibility of the Delta VOC $\psi \sim U(1.0, 3.0)$;
- the date of the introduction of the Delta VOC. We consider values between 45 days before and after the date when Delta was responsible for at least 5% of sequenced samples according to the data from Ref. 64;
- the IFR multiplier $\sim U(0.5, 2.0)$; this number multiplies the IFR from Ref. 38;

- the percentage of deaths reported ~ $U$ (1%, 100%). The model is calibrated separately for the different countries during the period 2020/10/01–2021/10/01. For each country, we run 20 iterations of the ABC-SMC algorithm using the weighted mean absolute percentage error (wMAPE) on real and simulated weekly deaths as distance metric. Full details on calibration technique and posterior distributions of free parameters are reported in the Supplementary Information.

## Reporting summary

Further information on research design is available in the Nature Portfolio Reporting Summary linked to this article.

## Data availability

The data used for this study is publicly available. Data on vaccinations can be downloaded at https://ourworldindata.org/covid-vaccinations, demographic data at https://population.un.org/wpp/Download/Metadata/Documentation/, mobility data at https://www.google.com/covid19/mobility/, and epidemiological data at https://github.com/CSSEGISandData/COVID-19.

## Code availability

All codes are available on Github at link https://github.com/ngozzi/vaccine-lmic and on Zenodo at Ref. 69.

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

## Acknowledgements

M.C. and A.V. were partially supported by the Bill & Melinda Gates Foundation (award number INV006010). N.E.D., M.E.H. and I.M.L. acknowledge support from NIH-R01 AI139761. M.E.H., I.M.L., and A.V. acknowledge support from the Ron Conway Family and the Emerson Collective. We acknowledge support from Google Cloud and Google Cloud Research Credits program. The content is solely the responsibility of the authors. N.G. acknowledges support from the DTA3/COFUND project funded by the European Union's Horizon 2020 research innovation programme under the Marie Skłodowska Curie Actions grant agreement No 801604. All authors thank the High-Performance Computing facilities at Greenwich University.

## Author contributions

N.G., M.C., N.P. and A.V. designed the study. N.G. and N.P. analyzed the data. N.G. implemented and run the epidemic model. N.G., M.C., N.E.D., I.M.L., M.E.H., N.P., and A.V. interpreted the results, wrote and approved the manuscript.

## Competing interests

The authors declare no competing interests.
