## [Peer Review File · Nature Communications]

Estimating the impact of COVID-19 vaccine inequities: a modeling studyREVIEWER COMMENTS

Reviewer #1 (Remarks to the Author):

I thought this was a very well-done and interesting paper on the potential of a more equitable vaccination distribution effort worldwide, nice job to the authors. This paper has clear potential interest to a wider audience, and accordingly my comments regard a couple clarification notes on the methods, and some suggestions on how to make this paper potentially more immediately approachable by a wide range of readers. I've broken these into major and minor comments, but I don't think any of these comments require changes to the model or redoing any analyses, more reframing and adding context for some parts of the paper.

Major comments:

- The paper takes the approach of addressing "allocation inequities", but the analyses regard vaccination rates above those that occurred--that is, the counterfactuals don't consider where the actual vaccines would have come from. There were conceivably ways for more vaccines to have been made available early on in the vaccination campaign (notably, perhaps more open vaccine development and manufacture), but I think it's worth noting that this doesn't explicitly address redistribution of the vaccines that we had, more what would have happened if the world had more vaccines, more quickly. Practically speaking, I don't think it would be in the scope of this paper to add the "redistribution" aspect, but I do think it's worth noting in the Discussion/Intro that there may have been ways to increase vaccine production faster.
- In terms of presentation, I think it would help to include the LMIC status/regional status of the different countries in Figures 2 and 4.. Maybe coloring the bars based on low/lower middle/middle income status, and then grouping based on WHO region? The first figure regards entirely the income status of different countries, and it would help the other figures be more coherent to continue this theme throughout.
- In this model, Delta largely influenced the transmissibility of disease, from what I understand. Various studies have also found there was a significant impact on severity as well ([https://www.thelancet.com/journals/laninf/article/PIIS1473-3099\(21\)00475-8/fulltext](https://www.thelancet.com/journals/laninf/article/PIIS1473-3099(21)00475-8/fulltext)), which would impact the potential averted deaths by vaccinating more before Delta's emergence. I don't think it's worth accounting for this in the model, but it is a further effect of Delta's introduction (and other variants, including Alpha, which was broadly suspected to be more virulent but not a great deal more transmissible). Notably, the reference used for IFR (Verity et al 2020) was from the first phase of the pandemic and only considered the original strain.

Minor comments:

- Were there any significant differences in age-specific contact rates between countries? Might be worth adding the specific contact matrices for the countries considered in the Supplementary Information.
- Figure 4B needs a legend on the color bar ("% of deaths averted")
- I assume this was the case, but when estimating the initial R_t , was the contact rate reduction from Google data at the time accounted for?

Reviewer #2 (Remarks to the Author):

Review of "Estimating the impact of COVID-19 vaccine allocation inequities: a modelling study"

In this manuscript, the authors fit a dynamic transmission model to COVID-19 epidemic data in 20 LMIC countries and estimate how many deaths could be averted had these countries had more vaccines.

Unfortunately, I don't think this study is of a publishable standard. In my opinion, there is certainly a lot of nice work here which the authors could develop, but I think the necessary changes would be fairly substantial.

As a side note, modelling studies should not be submitted for review without including code.

Major issues

1. The study is framed as an estimation of the impact of COVID-19 vaccine allocation inequities, but I don't think it explores the question of equity in a useful way. The authors show that lives would have been saved in LMICs if there had been more vaccine doses available or if doses had been available sooner, but this would be true for any country. The relevant question for equity here is: what would have been the effect of taking vaccine doses away from HICs and reallocating them to LMICs? If the authors had addressed that question, then their analysis could provide evidence in support of equity. Instead, their analysis provides evidence in support of faster vaccine manufacturing. That's fine, but the paper would need to be framed as such, and since speeding up vaccine manufacturing is a very different kind of problem than distributing the vaccines that were developed in an equitable manner, the resulting paper would probably be incomplete without an analysis of what it would take to achieve this faster manufacturing.

2. Some of the values for the proportion of deaths averted (i.e. up to 99%) are just unbelievable. And indeed, in Fig. 5 of the supplementary material, we see that the three countries with the highest proportion of deaths averted in the counterfactual scenarios — Afghanistan, Zambia, and Côte d'Ivoire — had early waves of deaths that the model is completely unable to capture. So these high proportions are clearly due to a flaw in model fitting, and are really not well supported at all. Moreover, the model itself makes no account for initially low ascertainment of deaths which later improved with time, which definitely happened in these countries and which will probably bias the model against accurately estimating the proportion of vaccine-avertible deaths in all countries (not just those with an uncaptured first wave).

Additional issues

"twenty LMIC sampled from all WHO regions" – how were these twenty LMICs chosen? This is critical information which needs to be stated in an obvious place. The word "sampled" usually implies some kind of random and blinded selection process, so I would recommend choosing a different word if this is not how the countries were chosen.

Results, first section – the first section of the results contains a lot of extraneous detail relating to Figure 1, which is really a sort of visual presentation of some background information. Generally speaking, it's a nice figure, but its function is essentially to set the scene so I think we can move past it a bit more quickly. It's not clear at first reading that the data sources described in the first paragraph of this section are just going to be used for this introductory figure, rather than for the main modelling analysis, so that also adds some confusion.

In figure 1, the size of the bubbles is not interpretable because no scale is provided. Also, smoothing over the distribution of first-vaccination dates across countries stratified by income level is not warranted when there are only 200-odd countries in the world – the data should be shown in bins or as individual points.

The sentence beginning with "As of October 1st 2022" is confusing – if 77% of individuals who completed the initial COVID-19 vaccination course live in high/upper middle income countries, then how could 50% of them live in LMICs? The other number here should be 23%... is the upper middle income country group being included in both the 77% and the 50%? Not sure what is going on here.

Second section – "SLIR" is pretty non-standard, please describe this as SEIR. It's fine to call the latent compartment L instead of E, but this is still generally called SEIR.

"the model takes as input... non-pharmaceutical interventions" – I would argue that the model takes as input mobility, which is obviously related to NPIs but is not the same as NPIs.

Third section (Counterfactual vaccination scenarios)

"US-equivalent vaccination" – this is a bit confusing since leading up to this point the paper seems to say that a scenario that is reflective of high-income countries generally will be looked at. I don't

think the alternative US-like, EU-like, and Israel-like vaccination scenarios add anything to the paper. I think the authors should just pick one (if they are going to stick with this methodology).

Fourth section (Estimate of NPIs)

How exactly the time series of mobility is changed to result in an extending of restrictions by X weeks is never fully described, as far as I can see, and it is not obvious.

Moreover, an X% increase in NPIs is a bit vague. I think this is more transparently described as a X% decrease in mobility, because at least then it is clear what is being measured.

Methods

“Indeed, the number of contacts scale with the square of the number of individuals” — this requires empirical support. OK, yes, if people are behaving like molecules leaving their homes at random times and bumping into each other, then the number of contacts would scale with the square of mobility, but this is not necessarily how mobility works. For example, workplace mobility might drop by 40% because people no longer go to the office on Mondays and Fridays, but on the three remaining days they still encounter each other at the same rate (which would make the number of contacts linear with mobility). Or, people go to bars and restaurants less, but when they do, they still sit with the same number of friends and interact with the same number of staff, which provide the main risk of infection (again, this would be a linear scaling).

Reviewer #3 (Remarks to the Author):

Gozzi and colleagues provide a very useful and well-conceived simulation setting to evaluate the effects of access to the COVID-19 vaccine due to inequities in the lower middle- and low-income countries. They explore counterfactual scenarios where they assume the same per capita daily vaccination rate reported in selected high-income countries. In the absence of equitable allocation, they also estimate the amount of additional effort necessary to offset vaccine shortage by means of non-pharmaceutical interventions.

The manuscript is well-written, and I enjoyed reading it very much. My reactions follow next:

Major issues to consider:

i) Counterfactual vaccination scenarios are constructed in which the authors estimate the impact of the counterfactual vaccine rollout as the percent reduction in deaths compared to the actual vaccine rollout. This calculation is based on a model that assumes that the vaccine mechanisms of action are efficacious against infection and death. Then, independent models are fit to each country yielding country-specific parameter estimates. In this context, country-specific estimates of infection incidence are indirectly affected by the vaccination history in the country and do not reflect the baseline incidence rate that would have been obtained in the country in the absence of vaccination, let's say λ_0 . I understand that this latter estimate is a necessary input to the simulations under the counterfactual scenarios. It is my understanding, and I might be wrong, that baseline incidences used in the counterfactual simulations are in fact lower than λ_0 . If my reasoning is right, estimates of deaths averted would be higher than the ones described. Please clarify. Notice that I am not asking for additional simulations/model fitting. I am just confused about how “strictly counterfactual” are the counterfactual scenarios presented.

ii) “Data to estimate the impact of NPIs on transmission dynamics come from the COVID-19 Community Mobility Report”. Although NPIs motivated by restraining mobility played an important role in the early times of the epidemics, the use of masks motivates current strategies. In fact, simulation results hint that infection transmission could be contained by the use of masks alone (see for example <https://doi.org/10.1016/j.idm.2020.04.001>). Since social distance-based intervention strategies imply important economic losses, NPIs based on the use of masks allow for more feasible and cost-effective comparisons with vaccines. I would like to see at least a few sentences on how mask-based NPIs score against mobility-based NPIs and vaccines.

Minor comments:

i) Technical issues regarding model fitting strategies such as parameter identifiability, convergence of numerical procedures, and assessment of goodness of fit need to be addressed in the supplemental material.

ii) As a third alternative to full vaccination and NPIs, delaying the second vaccine dose has been proposed, in the recent past, as a means to increase the number of individuals with partial protection. The two alternatives mentioned in this review, use of masks and vaccine schedule delays, might compare favorably against NPIs targeting mobility given the economic impact of the latter. The authors might care to comment on their motivations to present simulations focusing solely on mobility-based NPIs if they agree that such a discussion might entertain the reader's interest. I am certainly curious about their motivations.

iii) Section Results, 2nd paragraph, pg 2: "... vaccination level proposed by WHO >>has<< an interim target by the end of 2021" – has -> as

iv) Figure 2 B: If I understand it correctly, I would expect just one color hue above the dashed red line. This is the case for the Philippines and Ghana but not Pakistan, why? Also, explain in the figure caption the meaning of the white circle.

v) Pg 8: "This result, combined with the difficulty of implementing additional NPIs >>is<< these settings, underlines the largely..." is -> in

Detailed response to Reviewer #1

We thank the reviewer for their comments and suggestions. We have given these comments thoughtful consideration and made, where appropriate, the requested revisions to the manuscript.

I thought this was a very well-done and interesting paper on the potential of a more equitable vaccination distribution effort worldwide, nice job to the authors. This paper has clear potential interest to a wider audience, and accordingly my comments regard a couple clarification notes on the methods, and some suggestions on how to make this paper potentially more immediately approachable by a wide range of readers. I've broken these into major and minor comments, but I don't think any of these comments require changes to the model or redoing any analyses, more reframing and adding context for some parts of the paper.

We are honoured that the reviewer found our work well executed and of potential wide interest.

Major comments:

- 1) The paper takes the approach of addressing "allocation inequities", but the analyses regard vaccination rates above those that occurred--that is, the counterfactuals don't consider where the actual vaccines would have come from. There were conceivably ways for more vaccines to have been made available early on in the vaccination campaign (notably, perhaps more open vaccine development and manufacture), but I think it's worth noting that this doesn't explicitly address redistribution of the vaccines that we had, more what would have happened if the world had more vaccines, more quickly. Practically speaking, I don't think it would be in the scope of this paper to add the "redistribution" aspect, but I do think it's worth noting in the Discussion/Intro that there may have been ways to increase vaccine production faster.

We thank the reviewer for raising this concern, which is also shared by reviewer 2. As the reviewer correctly noted, our focus is not directed towards the global redistribution of doses. Rather, our aim is to highlight the largely untapped potential of vaccines in lower middle and low-income countries. Nonetheless, it is important to note that during the initial months of vaccine rollouts, numerous doses were either left unused (due to factors such as vaccine hesitancy or logistical challenges) or discarded (due to expiry dates) [1, 2]. Estimates reported by the Financial Times based on the work Airfinity (a health data group) suggest that, by the start of 2022, around 300 million doses may have gone wasted [3]. More generally, the overall number of doses available at any given time is influenced by many variables such as production choices, supply chain issues, phase of the pandemic, details of vaccination protocols such as the time between first and second doses (on this point a recent paper, considering data from the UK, appeared in Nature Communications [4]), national/international policies among others. The approaches to set these variables not only could have been different, but they have been different during the COVID-19 pandemic. As we anticipate future global health emergencies, counterfactual analyses like the one we propose may offer valuable quantitative insights into the impact of unequal vaccine access, thereby informing more effective strategies for future public health responses.

To clarify that the modelling framework proposed does not explicitly explore re-allocation strategies nor restrict the global number of doses, we have modified the title of the paper and its main text accordingly. Furthermore, we have added a point about this in the limitations as well as a reflection on the points raised above.

[1] <https://www.bmj.com/content/374/bmj.n1893>

[2] <https://gh.bmj.com/content/7/4/e009010.full>

[3] <https://www.ft.com/content/b2267d3a-ef24-4f96-9c02-7a057d80b3e6>

[4] <https://www.nature.com/articles/s41467-023-35943-0>

- 2) In terms of presentation, I think it would help to include the LMIC status/regional status of the different countries in Figures 2 and 4. Maybe coloring the bars based on low/lower middle/middle income status, and then grouping based on WHO region? The first figure regards entirely the income status of different countries, and it would help the other figures be more coherent to continue this theme throughout.

We thank the reviewer for this comment. We acknowledge that the proposed layout of the figures was not helping the narrative. Therefore, following the suggestion, we have revised Figures 2-4. Now countries are divided according to the 6 WHO regions and coloured according to their income status (for consistency, the colour code used is the same as Figure 1).

- 3) In this model, Delta largely influenced the transmissibility of disease, from what I understand. Various studies have also found there was a significant impact on severity as well ([https://www.thelancet.com/journals/laninf/article/PIIS1473-3099\(21\)00475-8/fulltext](https://www.thelancet.com/journals/laninf/article/PIIS1473-3099(21)00475-8/fulltext)), which would impact the potential averted deaths by vaccinating more before Delta's emergence. I don't think it's worth accounting for this in the model, but it is a further effect of Delta's introduction (and other variants, including Alpha, which was broadly suspected to be more virulent but not a great deal more transmissible). Notably, the reference used for IFR (Verity et al 2020) was from the first phase of the pandemic and only considered the original strain.

We agree with the reviewer on this point. The Delta variant was indeed found to be more severe than previously circulating strains. Therefore, we added a note on this in the section dedicated to the model's description in the Supplementary Information (SI Section 2, pg 4). We are confident that this does not affect our main findings. Indeed, our modelling setup includes an IFR multiplier which is calibrated with other free parameters. As a consequence, while the baseline IFR is indeed taken from the Verity et al study on the ancestral strain, the actual IFR employed in the simulations can differ from this baseline according to this multiplier, thus accounting - at least in part - for the possible different severity of the Delta variant.

Minor comments:

- 1) Were there any significant differences in age-specific contact rates between countries? Might be worth adding the specific contact matrices for the countries considered in the Supplementary Information.

In the Supplementary Information, we have added a section reporting the distribution of individuals in different age groups, as well as the contact matrices of the different countries considered (SI section 9, pg 18). In terms of age distribution, the average age is 26 years, spanning from a minimum of 20.6 years in Uganda to a maximum of 34.8 years in Sri Lanka. There is, therefore, some variability in terms of population pyramid even if, as expected, the countries included in the study show a significantly younger population compared to high-income countries.

Upon examining the contact matrices, we recognize minor variations in contact patterns across the countries considered, with the highest contact rates consistently observed among younger age groups. To provide a more quantitative assessment of these differences in the context of infectious diseases, we also evaluate the spectral radius (i.e., maximum absolute eigenvalue) of the normalised contact matrices, taking into account the age distribution of each population across age groups. The basic reproductive number of an infectious disease characterised by an SEIR natural history, is proportional to the spectral radius of such matrices. Consequently, this metric represents the effective intensity of contacts relevant to the spread of an epidemic. We discover that the spectral radius ranges from a minimum of 12.95 in Angola to a maximum of 14.07 in Sri Lanka, indicating relatively minor overall differences in the countries considered in the study.

- 2) Figure 4B needs a legend on the color bar ("% of deaths averted")

We thank the reviewer for noticing this. We amended the figure in the main text.

- 3) I assume this was the case, but when estimating the initial R_t , was the contact rate reduction from Google data at the time accounted for?

The reviewer is correct, the initial R_t considers the mobility reduction from the Google Mobility Report.

Detailed response to Reviewer 2

We express our gratitude to the reviewer for their insightful comments and recommendations. We acknowledge the concerns they have raised and have made every effort to address each one thoroughly.

In this manuscript, the authors fit a dynamic transmission model to COVID-19 epidemic data in 20 LMIC countries and estimate how many deaths could be averted had these countries had more vaccines.

Unfortunately, I don't think this study is of a publishable standard. In my opinion, there is certainly a lot of nice work here which the authors could develop, but I think the necessary changes would be fairly substantial.

We respectfully differ from the reviewer's perspective, as we are confident that our work meets the standards for publication. Regardless, we have carefully considered and addressed the concerns raised by the reviewer, which have contributed to certainly strengthen and clarify the results of our work.

As a side note, modelling studies should not be submitted for review without including code.

We would like to direct the reviewer to the link to the code that was included in the original version of the submitted manuscript. The code can be accessed at the following URL: <https://github.com/ngozzi/vaccine-lmic>

Major issues:

- 1) The study is framed as an estimation of the impact of COVID-19 vaccine allocation inequities, but I don't think it explores the question of equity in a useful way. The authors show that lives would have been saved in LMICs if there had been more vaccine doses available or if doses had been available sooner, but this would be true for any country. The relevant question for equity here is: what would have been the effect of taking vaccine doses away from HICs and reallocating them to LMICs? If the authors had addressed that question, then their analysis could provide evidence in support of equity. Instead, their analysis provides evidence in support of faster vaccine manufacturing. That's fine, but the paper would need to be framed as such, and since speeding up vaccine manufacturing is a very different kind of problem than distributing the vaccines that were developed in an equitable manner, the resulting paper would probably be incomplete without an analysis of what it would take to achieve this faster manufacturing.

We appreciate the reviewer for highlighting this issue, which has also been brought up by Reviewer #1. Our goal though is not to examine or evaluate methods for varying global distributions of vaccine doses. Instead, our goal is to quantitatively investigate the substantial, yet largely untapped, benefits of vaccines in lower-middle and low-income countries (LMICs) under various counterfactual scenarios. Although it may seem intuitive that increased vaccine doses would yield benefits, our findings reveal that these advantages are quite heterogeneous across countries and dependent on numerous features of the vaccination campaign, such as the timing and the rate of the rollout. The proposed model provides a tool to explore the interplay and effects of these variables.

We believe that the value of this type of analysis lies in demonstrating the potential effects of vaccines in these regions, regardless of the number of lives protected by vaccines in other parts of the world. While our findings cannot be used to inform (re)distribution strategies directly, they contribute to the relatively limited body of research exploring the outcomes of various sharing and production

policies. Indeed, the total number of available vaccine doses at any given time is influenced by numerous factors, such as production decisions, the pandemic phase, supply chain issues, vaccination protocol specifics (e.g., the interval between first and second doses, as discussed in a recent Nature Communications paper using UK data [1]), and national/international policies, among others.

Furthermore, it is worth noting that even during the early months of vaccine rollouts, a significant number of doses went unused (due to factors such as vaccine hesitancy and logistical challenges) or were discarded (due to expiration dates) [2, 3]. Financial Times estimates, based on data from health analytics group Airfinity [4], suggest that by the beginning of 2022, approximately 300 million doses had been wasted. Clearly, additional research is needed to explore how the global distribution could have been optimized to minimize such waste.

As we anticipate future global health emergencies, counterfactual analyses like the one we propose may offer valuable quantitative insights into the impact of unequal vaccine access thereby informing more effective strategies for future public health responses.

We acknowledge that the initial presentation of our paper may not have clearly conveyed our intended objectives. Consequently, we have revised the text to emphasize that the proposed modeling framework does not explicitly consider re-allocation strategies or the total number of globally available doses. Additionally, we have included a discussion on this topic in the limitations section, a discussion on the points mentioned earlier, and modified the title of the paper removing the word “allocation”.

[1] <https://www.nature.com/articles/s41467-023-35943-0>

[2] <https://www.bmj.com/content/374/bmj.n1893>

[3] <https://gh.bmj.com/content/7/4/e009010.full>

[4] <https://www.ft.com/content/b2267d3a-ef24-4f96-gc02-7a057d80b3e6>

- 2) Some of the values for the proportion of deaths averted (i.e. up to 99%) are just unbelievable. And indeed, in Fig. 5 of the supplementary material, we see that the three countries with the highest proportion of deaths averted in the counterfactual scenarios — Afghanistan, Zambia, and Côte d'Ivoire — had early waves of deaths that the model is completely unable to capture. So these high proportions are clearly due to a flaw in model fitting, and are really not well supported at all. Moreover, the model itself makes no account for initially low ascertainment of deaths which later improved with time, which definitely happened in these countries and which will probably bias the model against accurately estimating the proportion of vaccine-avertible deaths in all countries (not just those with an uncaptured first wave).

We are grateful to the reviewer for this technical comment.

The model was indeed not accurately capturing deaths for a short early time span in a few countries, most visibly in Zambia and Cote d'Ivoire, as indicated by the reviewer. To address this issue, we have introduced a new, more advanced fitting technique and recalibrated the model for all countries. The updated calibration method, called Approximate Bayesian Computation Sequential Monte Carlo (ABC-SMC), is a natural extension of the technique we previously employed. We have provided

comprehensive details about the implementation of ABC-SMC in the Supplementary Information (SI section 3, pg 4).

The approach comprises a sequence of simple acceptance/rejection steps that become progressively more stringent. This enables us to begin with high error tolerances and broad prior distributions, gradually narrowing our focus to explore the most relevant regions of the parameter phase space with greater precision. In addition to the new calibration technique, we have adjusted the calibration start date to 2020/10/01 (instead of 2020/12/01) for all countries to account for the delay in deaths relative to infections, incorporating a two-month burn-in period. These modifications successfully address the issues highlighted by the reviewer.

Overall, we are pleased to report that, based on the estimates from the recalibrated models, our findings remain largely consistent across all scenarios. Concerning the country explicitly mentioned by the reviewer, Zambia, the averted deaths, when US vaccine availability is applied (first counterfactual scenario), go from 11.5K (96.7% if expressed as a relative variation with respect to the baseline, which considers the actual vaccine availability) to 12.9K (71.7%). When the US start date is applied to the actual vaccine administration (second counterfactual scenario) the averted deaths go from 3.4K (28.2%) to 4.0K (22.1%). In the case of Cote d'Ivoire, instead, they go from 14.1K (99.5%) to 15.3K (66.0%) averted deaths in the first scenario, and from 4.5K (32.5%) to 3.8K (16.3%) in the second. The figures expressed as relative percentage changes do vary, primarily because they now reference baseline simulations that account for the earlier waves that were previously missed (resulting in higher baseline simulated deaths). However, we emphasize that the absolute numbers remain quite similar.

With regards to the time-varying ability of countries to ascertain deaths, our initial decision not to model it explicitly was based on two main factors. First, we aimed to avoid increasing the model's complexity by introducing additional parameters. Second, we observed that the period initially considered (2020-12-01 to 2021-10-01) was relatively short compared to the entire pandemic and did not include the earliest months of the emergency, which were characterized by greater uncertainty and underreporting compared to later stages. Our assumption was that any variations in the countries' ability to ascertain deaths during our focus period would likely be minimal and would not significantly impact the validity of our main findings.

However, for the sake of completeness and in response to the reviewer's suggestion, we have developed an additional model that incorporates two parameters to describe a country's ability to detect deaths: one for the first half and one for the second half of the simulation period. Comprehensive details about the implementation can be found in the Supplementary Information (SI Section 8, pg 15). We use the 95% confidence interval of the posterior for the first underreporting factor, obtained from the model presented in the main text, to inform its prior distribution. The second underreporting parameter can vary within a range of +/- 20 percentage points relative to the first one (this percentage deviation is a new parameter that is calibrated in the ABC-SMC process).

In general, this model yields a higher value for the second reporting factor compared to the first one, indicating an improvement in death detection. However, these estimated variations are typically small. Additionally, the averted deaths estimated using this new model consistently fall within the interquartile range of the estimates obtained with the single death reporting parameter model, with very few exceptions where the new figures still fall within the 90% confidence interval. Despite its

simplicity, this secondary model demonstrates that our findings remain robust even when accounting for potential variations in the ability to detect deaths in the selected countries.

Additional issues:

- 1) “twenty LMIC sampled from all WHO regions” – how were these twenty LMICs chosen? This is critical information which needs to be stated in an obvious place. The word “sampled” usually implies some kind of random and blinded selection process, so I would recommend choosing a different word if this is not how the countries were chosen.

We thank the reviewer for this question. We selected the countries to include representatives from all WHO regions based on data availability: deaths, infections, vaccinations, mobility, and Delta prevalence as guiding criteria. However, as the reviewer correctly highlights, we did not follow any random and blinded selection process. Therefore, following the suggestion, we replaced the word “sampled” with the more apt description “selected.”

- 2) Results, first section – the first section of the results contains a lot of extraneous detail relating to Figure 1, which is really a sort of visual presentation of some background information. Generally speaking, it’s a nice figure, but its function is essentially to set the scene so I think we can move past it a bit more quickly. It’s not clear at first reading that the data sources described in the first paragraph of this section are just going to be used for this introductory figure, rather than for the main modelling analysis, so that also adds some confusion.

We thank the reviewer for raising this point. In the first section of the results, as noted by the reviewer, we tried to set the stage for what comes later, providing an analysis of different variables that can be used to quantify disparities in vaccine rollouts across income levels. The text provides a general description of what we believe to be the key take home messages from the figure.

Although some of the data displayed in the figure is not utilized later in the analysis, certain aspects are crucial for the model, particularly the timeline of vaccinations in terms of doses and their start dates. We have revised the text to emphasize which data is employed in the model.

- 3) In figure 1, the size of the bubbles is not interpretable because no scale is provided. Also, smoothing over the distribution of first-vaccination dates across countries stratified by income level is not warranted when there are only 200-odd countries in the world – the data should be shown in bins or as individual points.

We thank the reviewer for highlighting the lack of scale in figure 1. In order to make the figure interpretable from a quantitative standpoint, we added the minimum and maximum values related to bubble sizes to the legend of Fig. 1B. We also amended Fig. 1C using discrete bins rather than a smoothed distribution.

- 4) The sentence beginning with “As of October 1st 2022” is confusing — if 77% of individuals who completed the initial COVID-19 vaccination course live in high/upper middle income

countries, then how could 50% of them live in LMICs? The other number here should be 23%... is the upper middle income country group being included in both the 77% and the 50%? Not sure what is going on here.

We thank the reviewer for flagging this inconsistency, the sentence is indeed wrong as reported. What we meant is that 77% of the people living in high and upper middle income countries completed the initial vaccination cycle as of October 1st 2022, while only 50% of those living in LMIC had received the same treatment. In the revised text we corrected the sentence.

- 5) Second section — “SLIR” is pretty non-standard, please describe this as SEIR. It’s fine to call the latent compartment L instead of E, but this is still generally called SEIR.

We thank the reviewer for this suggestion. We changed the reference to the compartmental setup from “SLIR” to “SEIR” throughout the text.

- 6) “the model takes as input... non-pharmaceutical interventions” – I would argue that the model takes as input mobility, which is obviously related to NPIs but is not the same as NPIs.

We thank the reviewer for raising this point. Indeed, we use mobility data that can act only as a proxy for NPIs adoption. Therefore, in the revised text, we opted for the formula “the model takes as input... proxy data for NPIs”.

- 7) “US-equivalent vaccination” — this is a bit confusing since leading up to this point the paper seems to say that a scenario that is reflective of high-income countries generally will be looked at. I don’t think the alternative US-like, EU-like, and Israel-like vaccination scenarios add anything to the paper. I think the authors should just pick one (if they are going to stick with this methodology).

We thank the reviewer for this comment. In the initial draft of the paper, we had the three cases shown in the main text. However, after some thinking we came to a realisation similar to the thought of the reviewer and opted to show only one in the main text as the overall message coming from the different cases is indeed similar. Nevertheless, we believe it is important to keep the others in the Supplementary Information as sensitivity analysis to slightly different scenarios.

- 8) How exactly the time series of mobility is changed to result in an extending of restrictions by X weeks is never fully described, as far as I can see, and it is not obvious.

This is explained in section 6.3 of the Supplementary Information. Technically, in the new simulations with X% additional NPIs we modify the contact reduction factor as $c'(t) = c(t)(1 - X/100)$. Hence, we assume that the transmissibility reduces by a further X%. This transformation is applied to contact reduction factors after week 51 of 2020 (taken as the start of vaccinations in the US) and sustained for n weeks (where n can vary from 4 to 40). As a concrete example, if n=4, we will apply the transformation just described to the 4 weeks following reference week 51 of 2020. We further clarified this point both in the main text and in the Supplementary Information (main pg 6, SI section 6.3 pg 11).

- 9) Moreover, an X% increase in NPIs is a bit vague. I think this is more transparently described as a X% decrease in mobility, because at least then it is clear what is being measured.

We thank the reviewer for this comment. Following a similar remark from Reviewer 3, we have clarified the description of this counterfactual scenario.

As was explained in section 6.3 of the Supplementary information, in this counterfactual scenario we increase the NPIs by modifying the effective contact reduction factor as

$$c'(t) = c(t) * d$$

Where $d = (1 - X/100)$. In the expression, the factor $c(t)$ is the contact reduction parameter estimated via mobility data. The factor d instead describes the strengthening of NPIs that would bring an overall further reduction in transmissibility of X%. As such, it can be thought of as a mix of supplementary measures that include mobility reduction, social distancing, but also increased adoption of face masks.

We have changed the text and the relevant figure to clarify this point. We hope that these new changes, stimulated by the reviewers' comments, improved the clarity of the paper (main pg 6, SI section 6.3 pg 11).

- 10) "Indeed, the number of contacts scale with the square of the number of individuals" — this requires empirical support. OK, yes, if people are behaving like molecules leaving their homes at random times and bumping into each other, then the number of contacts would scale with the square of mobility, but this is not necessarily how mobility works. For example, workplace mobility might drop by 40% because people no longer go to the office on Mondays and Fridays, but on the three remaining days they still encounter each other at the same rate (which would make the number of contacts linear with mobility). Or, people go to bars and restaurants less, but when they do, they still sit with the same number of friends and interact with the same number of staff, which provide the main risk of infection (again, this would be a linear scaling).

We thank the reviewer for raising this point. We agree that our assumptions do not capture the complexity of human interactions and mobility in full. However, our model operates at the country level. Arguably, the most conservative approximation, in absence of detailed mobility data, is to consider a quadratic dependence between contacts and mobility. This choice is also further supported by the documentation provided by Google LLC about their mobility reports which reads "The data shows how **visits** to places, such as grocery stores and parks, are changing in each geographic region" [1]. Therefore, data published by Google explicitly report variations in visits (i.e., "head counts" and not other definitions of mobility) to specific locations.

We added a clarification to this point in the methods and supplementary information (SI section 4 pg 8).

[1] https://www.google.com/covid19/mobility/data_documentation.html?hl=en

Detailed response to Reviewer 3

We are grateful to the reviewer for their comments and suggestions. We considered each comment carefully and revised the manuscript accordingly.

Gozzi and colleagues provide a very useful and well-conceived simulation setting to evaluate the effects of access to the COVID-19 vaccine due to inequities in the lower middle- and low-income countries. They explore counterfactual scenarios where they assume the same per capita daily vaccination rate reported in selected high-income countries. In the absence of equitable allocation, they also estimate the amount of additional effort necessary to offset vaccine shortage by means of non-pharmaceutical interventions.

The manuscript is well-written, and I enjoyed reading it very much. My reactions follow next:

We are glad that the reviewer enjoyed reading our work.

Major issues to consider:

- 1) Counterfactual vaccination scenarios are constructed in which the authors estimate the impact of the counterfactual vaccine rollout as the percent reduction in deaths compared to the actual vaccine rollout. This calculation is based on a model that assumes that the vaccine mechanisms of action are efficacious against infection and death. Then, independent models are fit to each country yielding country-specific parameter estimates. In this context, country-specific estimates of infection incidence are indirectly affected by the vaccination history in the country and do not reflect the baseline incidence rate that would have been obtained in the country in the absence of vaccination, let's say λ_0 . I understand that this latter estimate is a necessary input to the simulations under the counterfactual scenarios. It is my understanding, and I might be wrong, that baseline incidences used in the counterfactual simulations are in fact lower than λ_0 . If my reasoning is right, estimates of deaths averted would be higher than the ones described. Please clarify. Notice that I am not asking for additional simulations/model fitting. I am just confused about how "strictly counterfactual" are the counterfactual scenarios presented.

The reviewer is correct. Our baseline incidence (let's call it for simplicity " λ_1 ") is derived from simulations that employ the real vaccination history of each country. It is therefore lower than " λ_0 " which is the incidence that would have been observed in the lack of any vaccine dose (assuming, of course, that vaccines offer protection from infection). Nonetheless, our goal is to estimate the impact of vaccine inequities and not vaccine impact per se. Therefore, our baseline has to be a scenario with the actual unfolding of the vaccine rollout in each country and not one with a total lack of vaccines. Indeed, the question we are trying to answer is "How beneficial would a US vaccine availability have been in LMIC with respect to the observed LMICs vaccine availability?" rather than simply "What would the impact of US vaccine availability have been in LMICs?". We hope this

clarifies the doubts raised by the reviewer. We added a clarification in the main text and in the supplementary information to make this point clearer (main page 4).

- 2) "Data to estimate the impact of NPIs on transmission dynamics come from the COVID-19 Community Mobility Report". Although NPIs motivated by restraining mobility played an important role in the early times of the epidemics, the use of masks motivates current strategies. In fact, simulation results hint that infection transmission could be contained by the use of masks alone (see for example <https://doi.org/10.1016/j.idm.2020.04.001>). Since social distance-based intervention strategies imply important economic losses, NPIs based on the use of masks allow for more feasible and cost-effective comparisons with vaccines. I would like to see at least a few sentences on how mask-based NPIs score against mobility-based NPIs and vaccines.

We appreciate the reviewer's comment and recognize that our description of NPI modeling was just concerning the technical implementation. As a result, we have added further discussion in the revised text, as suggested by the reviewer. We chose to use the COVID-19 Community Mobility Report published by Google LLC for modeling NPIs because it offers detailed and standardized data that can be easily applied across various geographies and timelines as a proxy for the adoption of NPIs aimed at reducing transmission. Unfortunately, we do not have access to datasets with a similar level of detail regarding the adoption of face masks or other complementary forms of NPIs that do not necessarily involve reducing mobility. However, we would like to emphasize that mobility data has been widely and successfully employed to incorporate the impact of NPIs on COVID-19 evolution in epidemic models [1,2,3,4].

Furthermore, while it is true that the time-varying modulation of NPIs is modelled with Google mobility data, we also calibrate the initial R_t of the simulations. This implies that, even without time-varying parameters, we could consider the attenuating effect of face masks and other NPIs on transmissibility into the transmission rate. These points are now discussed in the limitations section of the model.

Following the reviewer's comment, we have also rephrased the way we present the counterfactual scenario 3 where we modify the timeline of NPIs to counterbalance the scarcity of vaccines. In this scenario (as described in the supplementary information of the first submission) we simply rescale the contact reduction factor as

$$c'(t) = c(t) * d$$

where $d = (1 - X/100)$. The factor $c'(t)$ directly multiplies the transmission rate in the calculation of the force of infection. $c(t)$ is the contact reduction estimated via proxy data from Google LLC. The factor d instead encodes the strengthening of NPIs. X is indeed the % increase we referred to. Hence, the factor d can be considered as a mix of supplementary measures devoted to further reducing transmissibility, such as masks mandates. We have changed the narrative to highlight this point. We thank the reviewer for raising this point as we believe that it helped us to improve the presentation of results (main pg 6, SI section 4 pg 8, SI section 6.3 pg 11).

- [1] <https://www.science.org/doi/10.1126/sciadv.abco764>
- [2] <https://www.nature.com/articles/s42254-021-00407-1>
- [3] <https://www.nature.com/articles/s41586-020-2923-3>
- [4] <https://www.science.org/doi/10.1126/science.abb4218>

Minor comments:

- 1) Technical issues regarding model fitting strategies such as parameter identifiability, convergence of numerical procedures, and assessment of goodness of fit need to be addressed in the supplemental material.

We thank the reviewer for this comment. We completely agree and indeed some of the details about the numerical procedures were described. In the revised version of the manuscript, we have added more information about the numerical simulations, and especially we added information about the goodness of the various fits computed using the weighted mean absolute percentage error (wMAPE) in the tables of the posterior distributions in the supplementary information.

Also following the comments from Reviewer #2, we have introduced a new, more advanced calibration technique. In the original submission, we employed the simple ABC rejection algorithm, selecting the top 1000 simulations (based on wMAPE) out of 1 million for each country. Indeed, defining a priori a threshold for all countries was quite challenging. Simultaneously, specifying custom thresholds for each country was not ideal. Consequently, we opted to allocate the same (large) computational budget for each country and evaluate the model's performance in each case by selecting the top N parameter sets as posterior.

Now we use a natural extension of the rejection algorithm based on Sequential Monte Carlo approaches (ABC-SMC). This technique solves our previous issues through a more solid and principled calibration. In short, this approach consists of a sequence of simple acceptance/rejection steps which are progressively less tolerant. In this way, we can start from high error tolerances and wide prior distributions and explore more and more accurately the interesting regions of the parameter phase space as we proceed. We note how, after repeating all the calibration and simulation efforts, the overall findings are confirmed, therefore our approach is robust to changes in the calibration step. Full details on our new approach are provided in the Supplementary Information (SI section 3 pg 4).

- 2) As a third alternative to full vaccination and NPIs, delaying the second vaccine dose has been proposed, in the recent past, as a means to increase the number of individuals with partial protection. The two alternatives mentioned in this review, use of masks and vaccine schedule delays, might compare favorably against NPIs targeting mobility given the economic impact of the latter. The authors might care to comment on their motivations to present simulations focusing solely on mobility-based NPIs if they agree that such a discussion might entertain the reader's interest. I am certainly curious about their motivations.

We thank the reviewer for raising this point. The referee is correct. As mentioned above, several countries such as the UK changed the recommended, and initially authorised, vaccination protocol thereby delaying second doses. As we recall, the decision was rather controversial, as it looked (to some) as a forced decision due to necessity rather than scientific evidence. However, a very recent analysis published in Nature Communications provides great support for it [1]. In light of this result, we agree with the reviewer that changes to vaccination protocols might be used, at least partially, to offset the lack of full courses of doses. We have added a discussion of this point in the main paper (main pg 6 section “Estimate of NPIs required to offset vaccine inequity”).

[1] <https://www.nature.com/articles/s41467-023-35943-0>

- 3) Section Results, 2nd paragraph, pg 2: “... vaccination level proposed by WHO >>has<< an interim target by the end of 2021” – has -> as

We thank the reviewer for spotting this error, we corrected it in the revised text.

- 4) Figure 2 B: If I understand it correctly, I would expect just one color hue above the dashed red line. This is the case for the Philippines and Ghana but not Pakistan, why? Also, explain in the figure caption the meaning of the white circle.

The dashed red line in Fig4B (we believe the reviewer mistakenly wrote Fig2B) indicates the level corresponding to the median % averted deaths with a US-equivalent vaccination rate, as indicated in the caption and in the main text. This quantity is represented, together with interquartile ranges, in Figure 2B. Here we see that the Philippines would have averted ~79% of deaths in this vaccine-rich scenario. Therefore, in Fig4B we see that the red dashed line in the contour plot for the Philippines correctly falls in the colour hue dedicated to the range 70%-80%, and it has only two colour hues above (the one for the ranges 80%-90% and 90%-100%). Very similar remarks can be done for the other two countries. Pakistan would have averted ~72% of deaths with US-equivalent vaccinations, and therefore the red dashed line in Fig4B falls in the colour hue of the range 70%-80%, and it has two colour hues above (those for ranges 80%-90% and 90%-100%). Ghana would have averted ~59%, and therefore the red dashed line in Fig4B lies in the colour hue related to the range 50%-60% and it has 4 colour hues above (ranges 60%-70%, 70%-80%, 80%-90%, 90%-100%). Following a remark from Reviewer 1 we added a label to the colour bar in order to make the reading of the plot easier. We also added a clarification on the white circle in the caption of Fig4. We hope this clarifies the reading of the figures.

- 5) Pg 8: “This result, combined with the difficulty of implementing additional NPIs >>is<< these settings, underlines the largely...” is -> in

Again, many thanks for spotting a mistake. We corrected it.

REVIEWERS' COMMENTS

Reviewer #1 (Remarks to the Author):

I'm happy with the revisions the authors made and feel this is a strong manuscript.

Reviewer #2 (Remarks to the Author):

Re-review of "Estimating the impact of COVID-19 vaccine inequities: a modeling study"

First, I would like to apologize for missing the code that was indeed shared with the previous version of the manuscript. I don't really know how I missed it, but I would like to assure the authors that I did read their manuscript thoroughly.

I stated before that the work was not of a publishable standard. This was due partly to a problem with model fitting which the authors have now corrected.

My other major criticism was that the paper doesn't address the issue of inequity per se, because it is not about redistributing the vaccine doses that were actually available.

The idea that rich countries should give up vaccine doses in the next pandemic may not be politically realistic, so I would understand an argument that it is more viable to have "fast vaccine access for everyone" as a goal. That is the scenario that this paper explores.

However, I don't think removing the word "allocation" from the title has addressed this issue. This paper is not about equity per se. It is about increasing availability of vaccines. Obviously, the result of that, as modelled here, is more equitable. But I still think the title, abstract, and aspects of the main text are not accurately representing what the paper is about.

The title and abstract both refer to the impact/effects of "COVID-19 vaccine inequities". The introduction then says the paper's aim is "quantifying the effect of a more equitable distribution of vaccines on COVID-19 mortality".

I really think the reasonable interpretation of this concept of an "equitable distribution of vaccines" or "vaccine equity" is that it pertains to evening out the distribution of vaccine doses, not increasing the amount of vaccine doses and evening out their distribution at the same time.

As for my more minor concerns, the authors have clarified many aspects of their manuscript substantially. I have a few remaining points.

First, in "Counterfactual vaccination scenarios", and in the captions of all figures reporting a percentage decrease in deaths, the authors should explicitly state that the percentage reduction in deaths they are quoting refers to the (as now updated) 2020/10/01 – 2021/10/01 period only. They state earlier in the Results that this is when the model is run, but it would still be possible to be measuring deaths averted over a different time period, such as since the start of the pandemic, so this should be made absolutely clear. Anyone who just glances at figures 2 or 3 is likely to draw the wrong conclusion.

Second, in all figures showing medians and confidence intervals / IQRs etc, the very fat and rounded ends of the confidence intervals are hard to interpret. For example, in Figure 2B, for Côte d'Ivoire, the upper end of the IQR seems to go just above 70%. But then in Table 7 of the supplement, it looks like the IQR goes up to 69%. I grant this is a minor difference, but the large rounded lines are an aesthetic choice, and I think in scientific communication, aesthetic choices should not sacrifice clarity.

Third, why is Uganda missing from Tables 6-8?

Reviewer #3 (Remarks to the Author):

My comments were appropriately addressed by the authors and I have no further questions.

Detailed response to reviewer 2

First, I would like to apologize for missing the code that was indeed shared with the previous version of the manuscript. I don't really know how I missed it, but I would like to assure the authors that I did read their manuscript thoroughly.

I stated before that the work was not of a publishable standard. This was due partly to a problem with model fitting which the authors have now corrected.

We would like to express again our gratitude to the reviewer for identifying potential issues with our previous fitting procedure. We truly appreciate their valuable feedback. We are delighted to note that the enhancements we have made in response were deemed overall satisfactory.

We have given to the remaining concerns a thoughtful consideration and made several changes to the manuscript. Please find below the detailed response to the points raised.

My other major criticism was that the paper doesn't address the issue of inequity per se, because it is not about redistributing the vaccine doses that were actually available.

The idea that rich countries should give up vaccine doses in the next pandemic may not be politically realistic, so I would understand an argument that it is more viable to have "fast vaccine access for everyone" as a goal. That is the scenario that this paper explores.

However, I don't think removing the word "allocation" from the title has addressed this issue. This paper is not about equity per se. It is about increasing availability of vaccines. Obviously, the result of that, as modelled here, is more equitable. But I still think the title, abstract, and aspects of the main text are not accurately representing what the paper is about.

The title and abstract both refer to the impact/effects of "COVID-19 vaccine inequities". The introduction then says the paper's aim is "quantifying the effect of a more equitable distribution of vaccines on COVID-19 mortality".

I really think the reasonable interpretation of this concept of an "equitable distribution of vaccines" or "vaccine equity" is that it pertains to evening out the distribution of vaccine doses, not increasing the amount of vaccine doses and evening out their distribution at the same time.

We thank the reviewer for raising this significant point of discussion. We acknowledge that our paper does not delve into the allocation problem. We mentioned this point explicitly in the main text in the first revision round, but we agree that it can be highlighted even further. To this end, we have taken additional steps to elucidate our goals and scope to avoid any confusion in this regard. We modified the abstract to highlight the angle of our analysis and added a few more clarifications in the main text.

We have chosen to retain the current title as we believe that the limited vaccine availability in LMICs was a direct result of inequities in the early access to COVID-19 vaccines. Consequently, we argue that our paper examines the effects of vaccine inequalities on the spread of COVID-19 in these countries during the initial stages of vaccine rollout, by, admittedly, considering scenarios with higher or earlier vaccine availability.

As for my more minor concerns, the authors have clarified many aspects of their manuscript substantially. I have a few remaining points.

First, in “Counterfactual vaccination scenarios”, and in the captions of all figures reporting a percentage decrease in deaths, the authors should explicitly state that the percentage reduction in deaths they are quoting refers to the (as now updated) 2020/10/01 – 2021/10/01 period only. They state earlier in the Results that this is when the model is run, but it would still be possible to be measuring deaths averted over a different time period, such as since the start of the pandemic, so this should be made absolutely clear. Anyone who just glances at figures 2 or 3 is likely to draw the wrong conclusion.

We agree with the reviewer on the significance of this aspect. Consequently, we have made revisions in the figures' legends and in the indicated section to ensure that they explicitly state the time frame considered for calculating the averted deaths.

Second, in all figures showing medians and confidence intervals / IQRs etc, the very fat and rounded ends of the confidence intervals are hard to interpret. For example, in Figure 2B, for Côte d'Ivoire, the upper end of the IQR seems to go just above 70%. But then in Table 7 of the supplement, it looks like the IQR goes up to 69%. I grant this is a minor difference, but the large rounded lines are an aesthetic choice, and I think in scientific communication, aesthetic choices should not sacrifice clarity.

We appreciate the reviewer for bringing this to our attention. Upon further investigation of the code, we discovered that the issue was not related to the rounding of the bar ends, but rather to the way *matplotlib* (the Python library used for visualisation) draws the corners of line segments. This became evident when we observed that the problem persisted even after removing the rounding. To rectify this discrepancy, we have successfully implemented a correction in all figures. As a result, the edges of the bars now precisely align with the estimated interquartile ranges. We thank the reviewer again for highlighting this issue, and we apologise for any confusion it may have caused.

Third, why is Uganda missing from Tables 6-8?

We thank the reviewer for spotting this, in the revised version we have added Uganda to Tables 6-8.